# Angiotensin Receptor 1 Blockers Prolong Time to Recurrence after Radiofrequency Ablation in Hepatocellular Carcinoma patients: A Retrospective Study

**DOI:** 10.3390/biomedicines8100399

**Published:** 2020-10-08

**Authors:** Antonio Facciorusso, Mohamed A. Abd El Aziz, Ivan Cincione, Ugo Vittorio Cea, Alessandro Germini, Stefano Granieri, Christian Cotsoglou, Rodolfo Sacco

**Affiliations:** 1Department of Medical Sciences, Gastroenterology Unit, Ospedali Riuniti di Foggia, 71122 Foggia, Italy; ugocea@yahoo.it (U.V.C.); saccorodolfo@hotmail.com (R.S.); 2Department of Surgery, Mayo Clinic, Rochester, MN 13400, USA; abdelmaksoud.mohamed@mayo.edu; 3Department of Clinical and Experimental Medicine, Faculty of Medical and Surgical Sciences, University of Foggia, 71122 Foggia, Italy; i.cincione@unifg.it; 4General Surgery Department, ASST-Vimercate, 20871 Vimercate, Italy; alessandro.germini@asst-vimercate.it (A.G.); stefano.granieri@asst-vimercate.it (S.G.); christian.cotsoglou@asst-vimercate.it (C.C.)

**Keywords:** sartans, liver cancer, HCC, RFA, survival

## Abstract

Inhibition of angiotensin II synthesis seems to decrease hepatocellular carcinoma recurrence after radical therapies; however, data on the adjuvant role of angiotensin II receptor 1 blockers (sartans) are still lacking. Aim of the study was to evaluate whether sartans delay time to recurrence and prolong overall survival in hepatocellular carcinoma patients after radiofrequency ablation. Data on 215 patients were reviewed. The study population was classified into three groups: 113 (52.5%) patients who received neither angiotensin-converting enzyme inhibitors nor sartans (group 1), 59 (27.4%) patients treated with angiotensin-converting enzyme inhibitors (group 2) and 43 (20.1%) patients treated with sartans (group 3). Survival outcomes were analyzed using Kaplan–Meier analysis and compared with log-rank test. In the whole study population, 85.6% of patients were in Child-Pugh A-class and 89.6% in Barcelona Clinic Liver Cancer A stage. Median maximum tumor diameter was 30 mm (10–40 mm) and alpha-fetoprotein was 25 (1.1–2100) IU/mL. No differences in baseline characteristics among the three groups were reported. Median overall survival was 48 months (42–51) in group 1, 51 months (42–88) in group 2, and 63 months (51–84) in group 3 (*p* = 0.15). Child-Pugh stage and Model for End-staging Liver Disease (MELD) score resulted as significant predictors of overall survival in multivariate analysis. Median time to recurrence was 33 months (24–35) in group 1, 41 (23–72) in group 2 and 51 months (42–88) in group 3 (*p* = 0.001). Number of nodules and anti-angiotensin treatment were confirmed as significant predictors of time to recurrence in multivariate analysis. Sartans significantly improved time to recurrence after radiofrequency ablation in hepatocellular carcinoma patients but did not improve overall survival.

## 1. Introduction

Despite the advancements in hepatocellular carcinoma (HCC) surveillance, chemoprevention (by decreasing the known risk factors), diagnostic tools, and therapeutic options, HCC still represents the third most common cause of cancer-related death worldwide and the leading cause of mortality amongst patients with liver cirrhosis [1,2].

Surgical resection or orthotopic liver transplantation (OLT) is considered the first-line option for early-stage HCC. However, in some cases, curative resection or OLT is not possible and, in that case, radiofrequency ablation (RFA) is considered to be a first-line treatment option [2,3]. Introduction of RFA as a treatment option achieved high success, for highly selected patients, with a five-year overall survival rate up to 70% [4,5,6]. However, the recurrence rate after RFA is still considered high (up to 80% within four years) [5,6]. Therefore, there is an unmet need for effective adjuvant therapy to decrease the recurrence after RFA. In that case, sorafenib as well as interferon have been investigated as an adjuvant therapeutic option. However, unfortunately, their long-term use was associated with severe adverse events and, consequently, a high withdrawal rate from the trials [7,8]. Moreover, their high cost as well as the lack of real-life clinical advantages represent further challenges against their adoption as a reliable adjuvant option [9,10]. Therefore, another option with low side effect profile and low cost is needed. In that case, theoretically, angiotensin-converting enzyme inhibitors (ACE-I) have anti-angiogenic and antifibrogenic activity; thus, they have been investigated as an adjuvant option for HCC. However, the results of their use as monotherapy were not satisfactory [11,12] and were associated with side effects such as cough and angioedema [11,13]. Despite this, data on the chemoprevention effect of angiotensin II type 1 receptor blockers such as sartans against HCC recurrence are still lacking. Pre-clinical studies seem to provide promising results and stand for a clear efficacy profile of sartans in HCC animal models [14] but data in humans are still lacking.

The aim of this study was to evaluate whether angiotensin II receptor 1 blockers can delay time to recurrence (TTR) and prolong overall survival (OS) when used in HCC patients after treatment with percutaneous RFA.

## 2. Materials and Methods

### 2.1. Patients

All adult patients treated with RFA as a first-line therapy for HCC at the University of Foggia between February 2004 and March 2015 were retrospectively reviewed. Indications to RFA treatment were: (1) HCC diagnosed by histology or by non-invasive criteria according to the American Association for the Study of Liver Disease (AASLD) guidelines [15]; (2) non-metastatic HCC patients in Barcelona Clinic Liver Cancer (BCLC) stage 0/A (single tumor nodule or up to three nodules of less than three centimeters), not suitable for surgical therapies; (3) preserved liver function (within Child-Pugh (CP) stage B7). The dataset 1 of the enrolled patients is presented in the Appendix A.

Contraindications to RFA were decompensated liver cirrhosis and at-risk tumor locations (superficial lesions adjacent to any part of the gastrointestinal tract). RFA was also avoided in case of nodules adjacent to hepatic vessels due to the risk of incomplete treatment because of the heatsink effect (i.e., heat loss by convection).

The study population was divided into three cohorts: (1) patients who did not receive ACE I or sartans (group 1); (2) patients with hypertension treated ACE I (group 2); (3) patients with hypertension treated with sartans (group 3).

Patients who had cross-overs from an anti-hypertensive drugclass to another were excluded from the analysis in order to avoid misclassification due to class switching.

Only patients under antihypertensive therapy for at least 2 years before RFA were included in group 2 and 3 to allow a reasonable induction period for the anti-proliferative and anti-angiogenic effects of such drugs.

This study was approved by our Institutional Review Board for retrospective evaluation of de-identified patients (n. 1426/20, approved by the Institutional Review Board of the Azienda Ospedaliera Ospedali Riuniti di Foggia on 10 July 2020). The database was locked in January 2019.

### 2.2. Treatment Protocol

The technical details of the ablative procedures performed in our center have been described before [6,16]. Briefly, all the procedures were performed under ultrasound guidance with a 150 W generator (Model 1500 L; RITA Medical System, Mountain View, CA, USA), connected to an expandable 15–14-gauge electrode with a 2.0 cm long exposed tip (expandable by means of seven hooks). After administration of analgesia (50 to 60 mg of propofol and 0.05 to 0.1 mg of fentanyl) as well as local anesthesia (5 to 15 mL of 1% lidocaine) by an anesthesiologist, an RFA needle was first inserted into the tumor. The electrode was placed into the center of the lesion maintaining the temperature of the needle tip at 80–110 °C for 10–12 min. After ablation, the needle was retracted while maintaining its tip hot in order to prevent thermal coagulation seeding or hemorrhage along the electrode track. For larger nodules, different applicator positions were adopted to create overlapping coagulation zones. For patients with multiple nodules, all lesions were treated in a single session. Every procedure was aimed at obtaining a 5 mm safety margin around the treated lesions. No antibiotic prophylaxis or anti-inflammatory drug was administered prior to therapy.

### 2.3. Patient Monitoring and Response Evaluation

Tumor response was assessed according to modified RECIST (mRECIST) criteria [17]. For analytical purposes, in the case of consecutive procedures, the best response achieved after the last RFA of the treatment series was considered.

Safety parameters were classified following the common terminology criteria for adverse events (CTCAE) 4.0 [18].

At recurrence, in case of intrahepatic disease, the elective treatment was RFA for single nodules and trans-arterial chemoembolization (TACE) for multifocal HCC, sorafenib (Nexavar^®^, Bayer, Leverkusen, Germany) if portal vein thrombosis or metastases occurred.

Pain and fever occurring after the procedure were managed individually. Clinical visits, including physical examination, laboratory analyses (transaminase, liver function panel, complete blood count) and serum alpha-fetoprotein (AFP), thoracoabdominal multi-phase CT-scan evaluation, and adverse events (AE) monitoring, were performed at the outpatient clinic 2 months after the procedure. In case of complete response, follow up visits were scheduled every 4–6 months. In the case of an incomplete response, a second treatment was planned in CP ≤ B7 patients.

### 2.4. Statistical Analysis

Categorical variables were reported as frequencies and percentages and continuous variables reported as medians and ranges or mean ± standard deviation as appropriate. Comparison of baseline parameters between the three study groups was performed using Kruskal–Wallis test for continuous variables and Chi-square test for categorical ones.

Overall survival (OS) and time to recurrence (TTR) were estimated from the date of the first RFA by Kaplan–Meier curves with plots and median (95% CI) and compared through the log-rank test.

The inferential analysis for time to event data, namely the factors influencing OS and TTR, was conducted using the Cox univariate and multivariate regression models to estimate hazard ratios (HR) and 95% CI. Statistically significant variables from the univariate Cox analysis were considered for the multivariate models.

In order to counteract the problem of multiple comparisons, since the study had more than two groups, the *p*-values obtained in Cox regression concerning the impact of anti-hypertensive drugclasses on the outcomes were corrected according to Bonferroni method [19,20].

The analysis was performed using R Statistical Software (Foundation for Statistical Computing, Vienna, Austria) and significance was established at the 0.05 level (two-sided).

## 3. Results

### 3.1. Clinical Characteristics

A total of 215 patients were included. Of them, 113 (53%) patients did not receive ACE I or sartans, 59 (27%) patients received ACE I and 43 (20%) patients received sartans. In group 3 (sartan group), 23 patients (53.4%) used losartan, 8 (18%) valsartan, 7 (16.2%) irbesartan and 5 (12.4%) olmesartan.

The main clinical features observed in the 215 patients who underwent RFA are shown in Table 1. There were no significant differences between the three study groups regarding patient and tumor characteristics. Median age of the whole population was 70 years (range 39–86), specifically 70 years (48–86) in patients not in antihypertensive therapy, 69 years (39–86) in patients treated with ACE I, and 72 years (48–82) in patients treated with sartans.In the whole cohort 171 patients (79.7%) were male, with 89 (79.5%), 45 (77.5%), and 37 (83.9%) male patients in the three groups, respectively. Median BMI was 23 (17–36) in the whole sample, with no difference among treatment groups (*p* = 0.45). Mean arterial pressure was 96.4 ± 15.5 mmHg, with slightly superior values in patients in group 2 (103.3 mmHg) and 3 (100 mmHg), as compared to the group not in antihypertensive treatment (87.6 mmHg; *p* = 0.09). Nearly 40% of patients presented mellitus diabetes, specifically 38 patients (34.2%) in group 1, 24 patients (40.8%) in group 2, and 19 patients (42%) in group 3 (*p* = 0.66). HCV-based cirrhosis was the most frequent etiology of the underlying liver disease (46.4% in the whole group, 42.4% in group 1, 51% in group 2, and 48.3% in group (3). There were 85% of patients in Child-Pugh stage A, with no difference among groups *(p* = 0.54). Model for End-Stage Liver Disease (MELD) score was 9 (6–17) without differences among groups (*p* = 0.48). Portal hypertension—defined by at least one of esophageal varices, platelet count <100,000/µL, and/or splenomegaly [21]—was diagnosed in 99 subjects (46.4%), in particular in 45 patients (40%) in group 1, 19 patients (32.6%) in group 2, and 31 patients (54.8%) in group 3 (*p* = 0.14). Median AFP was 24 IU/mL (1.1–2100), again with no difference among groups (*p* = 0.10). Most patients were in BCLC A stage (90.5% in group 1, 87.8% in group 2 and 90.4% in group 3, respectively; *p* = 0.88). Median number of nodules was 1 in all the study groups and in the whole cohort (*p* = 0.68). The median maximum tumor diameter in the three groups was 29 mm (14–45), 30 (10–45), and 30 (10–40), respectively (*p* = 0.92). Overall, patients had well-preserved performance status (Eastern Cooperative Oncology Group (ECOG) 0 in 100%).

Overall, 288 HCC nodules were treated, 87 (30.3%) ≤ 2cm and 201 (69.7%) between 2 and 5 cm.

### 3.2. Tumor Response and Safety Data

Objective response (complete response + partial response) was observed in 92.1% of patients. Specifically, 166/198 (84.4%) had a complete response and 15/198 (7.7%) had a partial response. Mean number of RFA sessions needed to achieve the objective response was 1.4 (±0.54) with a median time to response of 2 months (95% CI 1–3).

No treatment-related deaths were observed. Liver decompensation rate (assessed at 1 month) was 3.2% (7/215) while four patients (1.8%) experienced severe adverse events (AE grade 3/4), including one case of abdominal abscess and three cases of abdominal pain, from which patients recovered after hospitalization. No severe AEs related to anti-hypertensive drugs were observed.

### 3.3. Overall Survival

At a median follow up time of 108 months (95% CI: 89–188), 175 deaths were observed. Median OS was 58 months (49–72) and survival rate (SR) was 98.6%, 51.6% and 39.5% at 1, 4 and 5 years, respectively, in the whole cohort.

On univariate Cox regression analysis, age (HR 1.56, 1.01–2.42; *p* = 0.04)), CP score (HR 2.83, 1.70–4.71; *p* < 0.001), MELD score (HR 2.13, 1.21–3.75; *p* < 0.001), and AFP levels (HR 2.07, 1.89–3.5; *p* = 0.03) were predictors of OS (Table 2). The other variables tested in the univariate analysis for OS were not significant, specifically gender (HR 0.96, 0.54–1.69; *p* = 0.89), blood hypertension (HR 1.08, 0.79–1.34; *p* = 0.89), BMI (HR 1.25, 0.84–1.42; *p* = 0.46), mellitus diabetes (HR 1.34, 0.95–1.68; *p* = 0.25), etiology (HR for HCV 0.77, 0.46–1.41; HR for other etiology 1.54, 0.77–3.09), portal hypertension (HR 0.86, 0.56–1.30; *p =* 0.48), max diameter (HR 1.12, 0.68-1.82; *p* = 0.65), BCLC (HR 1.54, 0.80–2.98; *p* = 0.19), CLIP stage (*p* = 0.07), number of nodules (HR 1.75, 0.92–2.04; *p* = 0.25). Similarly, anti-angiotensin therapy was not found to be a significant predictor of OS (*p*=0.15); in particular, ACE I determined an HR for overall survival of 0.91 (0.76–2.10; *p* = 0.39) and sartans an HR of 0.71 (0.46–1.10; *p* = 0.12). Only CP stage (HR 2.58, 1.44–4.62; *p* = 0.001) and MELD score (HR 2.37, 1.33–4.22; *p* = 0.003) were confirmed as significant parameters on multivariate analysis (Table 2).On the other hand, age (HR 1.89, 0.43–2.44; *p* = 0.21) and AFP (HR 1.71, 0.45-3.2; *p* = 0.17) were not confirmed as predictors of OS in multivariate analysis.

Median OS was 48 months (42–51) in patients who did not receive either sartans or ACE-I therapy, 51 (42–88) in patients treated with ACE I and 63 months (51–84) in those under sartans (Figure 1; *p* = 0.15). No difference in terms of OS according to drug used (within the sartan class) was observed (*p* = 0.98).

### 3.4. Time to Recurrence

During the study follow-up, 153 patients experienced tumor recurrence, whereof 49 (32.3%) were intrahepatic local recurrences (i.e., in the same liver segment), 75 (49.4%) intrahepatic distant, and 29 (18.3%) consisted in extrahepatic metastatic spread. Median time to recurrence (TTR) was 33 months (31–42) with a recurrence-free survival (RFS) rate of 78.1%, 36.6%, and 33.3% at 1, 4, and 5 years, respectively.

Age (HR 2.02, 1.27–3.21; *p* = 0.002), number of nodules (HR 2.45, 1.32–3.04; *p* = 0.02), and type of anti-angiotensin therapy (*p* = 0.001) were predictors of TTR at univariate analysis. Specifically, ACE I determined an HR for TTR of 0.71 (0.46–1.12; *p* = 0.44), whereas only sartans determined a significant HR for delayed TTR of 0.49 (0.28–0.84; *p* = 0.03). The other parameters were not significantly associated with TTR, specifically gender (HR 1.63, 0.89–3.01; *p* = 0.07), blood hypertension (HR 1.28, 0.82–1.74; *p* = 0.79), BMI (HR 1.34, 0.89–1.78; *p* = 0.21), mellitus diabetes (HR 1.46, 0.91–2.04; *p* = 0.16), etiology (HR for HCV 0.92, 0.78–1.35, HR for other etiology 1.24, 0.87–3.89), CP score (HR 1.05, 0.57–1.92; *p* = 0.87), portal hypertension (HR 0.73, 0.47–1.14; *p* = 0.16), AFP (HR 1.10, 0.73–1.66; *p* = 0.62), MELD (HR 0.73, 0.43–1.24; *p* = 0.25), max tumor diameter (HR 0.90, 0.54–1.51; *p* = 0.71), BCLC (HR 1.11, 0.59–2.09; *p* = 0.72), CLIP stage (*p* = 0.15).

Only number of nodules (HR 1.45, 1.02–3.01; *p* = 0.04) and anti-angiotensin treatment (*p* = 0.008) were confirmed in multivariate setting (Table 3). In particular, while ACE I was not significantly associated with TTR (multivariate HR 0.78, 0.49–1.22; *p* = 0.31), sartans were confirmed as important predictors of delayed TTR in the multivariate setting (HR 0.47, 0.27–0.82; *p* = 0.009). On the other hand, age was not confirmed in multivariate analysis (HR 1.59, 0.73–2.34; *p* = 0.21).

Median TTR, when stratified by anti-angiotensin drug class, was 33 months (24–35) in group 1, 41 (23–72) in group 2, and 51 months (42–88) in group 3 (Figure 2; *p* = 0.001). Withingroup 3, there were no TTR differences with regard to drug molecule used (*p* = 0.87).

## 4. Discussion

Despite the recent improvement in surveillance programs and therapeutic strategy, HCC remains a major health problem. Even after radical treatments (such as surgery, liver transplantation, or RFA), tumor recurrence represents a challenge due to its high frequency and often more aggressive course. Therefore, in the last few years, there has been an increasing interest in adjuvant therapies aiming to decrease the tumor recurrence rate after primary therapy.

Despite the promising results of some retrospective reports and the theoretical advantages of sorafenib in the adjuvant setting, a broad multicenter randomized-controlled trial (Sorafenib as Adjuvant Treatment in the Prevention Of Recurrence of Hepatocellular Carcinoma (STORM)), enrolling 1114 HCC patients after resection or RFA, failed to find a significant improvement in RFS (primary endpoint) and OS [10,22,23]. This disappointing result was partly related to the high discontinuation rate of therapy because of severe adverse events or consent withdrawal [10]. Similarly, other drugs such as interferon, provided discordant results in the adjuvant setting due to their high cost and narrow therapeutic window [9,24].

Angiotensin II induces the release of vascular endothelial growth factor (VEGF) and enhances hepatic fibrosis through the production of transforming growth factor-beta 1 (TGF-β1) by Kupffer and activated stellate cells [14,25]. This aspect is particularly of interest in hepato-oncology, where the synergistic action of either pro-tumorigenic or profibrogenic properties of angiotensin II may play a pivotal role in hepatocarcinogenesis [12,14,25]. Therefore, it was not surprising that several studies conducted in different fields of human oncology suggest that ACE-I and sartans have antineoplastic properties [26,27,28]. However, despite these theoretical advantages of ACE-I, large clinical reports have found significantly decreased HCC recurrence rates only when these drugs were used in combination with other agents and, unfortunately, no difference in overall survival was observed in comparison to the control arm [11,13,29]. Indeed, most of the biological properties of angiotensin II, including the profibrogenic and pro-angiogenic activity, are mediated by receptor 1. Only in the last few years has angiotensin II receptor 2 been known to have specular effects as compared to receptor 1, by inhibiting cell proliferation and stimulating apoptosis in a variety of cell lines [30]. While ACE-I block upstream the activation cascade of the renin-angiotensin-aldosterone system (RAAS), thus preventing the binding of angiotensin II to both receptors, sartans selectively inhibit the activation of receptor 1. Moreover, it has been found that under sartan therapy, receptor 2 is overexpressed as a regulatory feedback mechanism to pharmacological inhibition [30]. Hence, the pro-apoptotic and anti-proliferative properties of angiotensin II are enhanced in patients treated with sartans. Furthermore, sartans do not predispose people to bothersome side effects (like cough) that, although not life-threatening, may represent a cause of treatment interruption in ACE-I treated patients. However, the role of the angiotensin II type 1 receptor blockers (namely sartans) remains poorly understood. Therefore, (and also based on the promising results of some animal studies [31]), we decided to retrospectively review data of our historical cohort of HCC patients treated with RFA to test whether sartans administered to hypertensive patients might be able to prolong survival and delay tumor recurrence. Primary indication of either sartanor ACE-I therapy was essential for hypertension, a very common condition in the Western world and often related to metabolic syndrome.

Our study represents the long-term follow-up of a previous report conducted at our Center aiming to investigate the effects of sartans on disease recurrence and survival in HCC patients after curative therapy [32].

Although OS was no different among the study cohorts (Figure 1), patients under sartan therapy showed significantly longer TTR in comparison to either patients under ACE-I and those not receiving any of the aforementioned drug classes (Figure 2). Our series confirmed the anti-tumor properties of ACE-I but found an even enhanced tumor-suppressing activity in patients treated with sartans, which supports the study hypothesis. Indeed, the anti-hypertensive drug class was shown, together with a number of tumor nodules as the sole factor at multivariate analysis, to be able to significantly influence TTR.

The post-hoc analysis, undertaken by means of Bonferroni correction of *p*-values, pointed out the efficacy of sartans since the significance threshold was reached only for group 3 (Table 3). Therefore, between the two classes of anti-hypertensive drugs, only sartan therapy resulted as a significant predictor of better TTR.

Portal hypertension is a well-known predictor of poor prognosis. Unfortunately, a direct measurement of portal vein pressure is not routinely practiced at our institution. However, we defined it based on indirect features which have been described before [21]. Interestingly, the beneficial antitumor activity was noted in the sartan group even though they tend to have more frequent portal hypertension. This, while being likely to underestimate the real presence of this condition (thus explaining the apparent discrepancy between the number of patients in CP A stage and the number of patients with portal hypertension in our study) represents a validated and reliable indicator of the hemodynamic imbalances occurring in a portal vein system in patients with liver cirrhosis [33]. Therefore, it seems that angiotensin II receptor-1 blockers showed a clear recurrence-free survival benefit, even in cases of more advanced underlying liver disease. This happened in spite of the theoretical concerns due to hemodynamic alterations occurring in patients with liver cirrhosis.

Although it is well known that as liver cirrhosis progresses the mean arterial pressure tends to decrease, in our study population patients did not show any contraindication to anti-hypertensive drugs throughout the study period. This could be because of the preserved liver function status registered at baseline.

Among angiotensin receptor blockers, telmisartan has been widely recognized to exert anti-proliferative effects in several kinds of cancers [34,35]. However, no patient in our series had been administered telmisartan, probably because of its higher dosage as compared to other sartans which may constitute a contraindication in patients affected by liver diseases. Consequently, and because no difference among single drugs was found regarding TTR and OS, the chemo-preventive activity of sartans against HCC recurrence seems to be class-related. However, the small sample size in our study doesn’t allow a definitive conclusion in such regard.

The aforementioned results might be of interest in oncology as sartans were found to determine interesting survival outcomes also in other kinds of cancers, for example renal carcinoma. A recent pooled analysis published by McKay et al.considering 4736 patients with metastatic renal cell carcinoma, of which 1487 were ACE-I/sartan users, demonstrated this positive association [36]. This therapy has also shown positive clinical outcomes in advanced stages (e.g., metastatic lung cancer) and highly aggressive tumor types (e.g., glioblastoma and pancreatic ductal adenocarcinoma), as suggested in a recent study by Liu et al. [37]. Similarly, the beneficial effects of sartans in cancer prevention were pointed out in several studies conducted in different kinds of malignancies. Two large epidemiological studies conducted in the UK [38] and in Taiwan [39] demonstrated a positive association between ACE-I/sartan use and a reduced risk of developing cancer, in particular colorectal cancer. Therefore, our study sheds further light on the potential anti-oncotic effects of sartans in clinical practice.

As expected, no severe AEs related to sartans were observed, thus confirming the excellent safety profile of these drugs. Thus, as described above, sartans seem to have an oncological benefit with an acceptable safety profile: they are inexpensive and widely available, which are both optimal characteristics for an adjuvant drug.

The current study has some weaknesses. First, its retrospective nature may have affected the reliability of the reported outcomes. However, as shown in Table 1, the three groups were highly comparable regarding baseline clinical and tumoral characteristics, thus obviating some of the selection bias. Moreover, the completeness of the data collection and adequate follow-up duration strongly support the robustness of our findings. Second, the small sample size (particularly with regard to the sartan group) requires further confirmation of our results in large cohort studies or randomized controlled trials. Furthermore, the low number of patients on sartan therapy may have limited our ability to reach an assertive conclusion regarding the subgroup analysis of OS and TTR stratified by drug molecule use. However, even in this small study sample, our series confirms the absence of drug-related differences of outcomes reported in other clinical reports analyzing angiotensin II-blocking agents [29].

Despite these limitations, particularly due to the retrospective nature of the analysis, our study is the first to report on the efficacy of sartans in increasing recurrence-free survival rates in HCC patients after radical therapy. We thus think that, due to their low cost and easy use, sartans may represent a promising therapeutic tool in the management of early HCC patients undergoing ablative or surgical therapies.

## Figures and Tables

**Figure 1 biomedicines-08-00399-f001:**
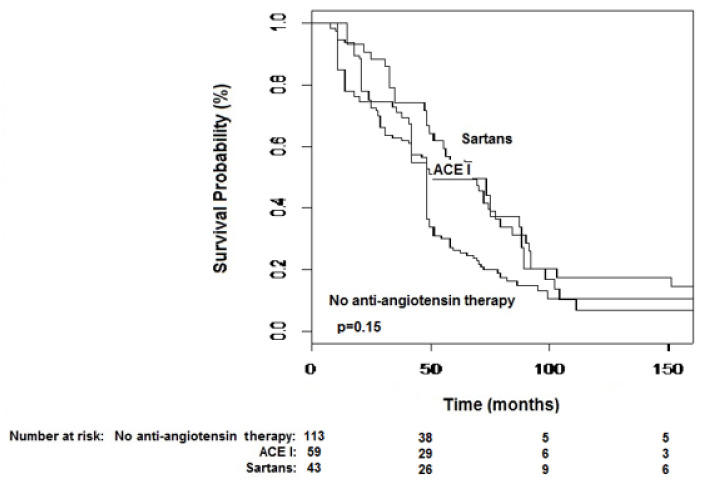
Kaplan–Meier curves of overall survival stratified by anti-hypertensive therapy. Median survival was 48 months (42–51) in group 1, 51 (42–88) in group 2 and 63 months (51–84) in group 3 (*p* = 0.15).

**Figure 2 biomedicines-08-00399-f002:**
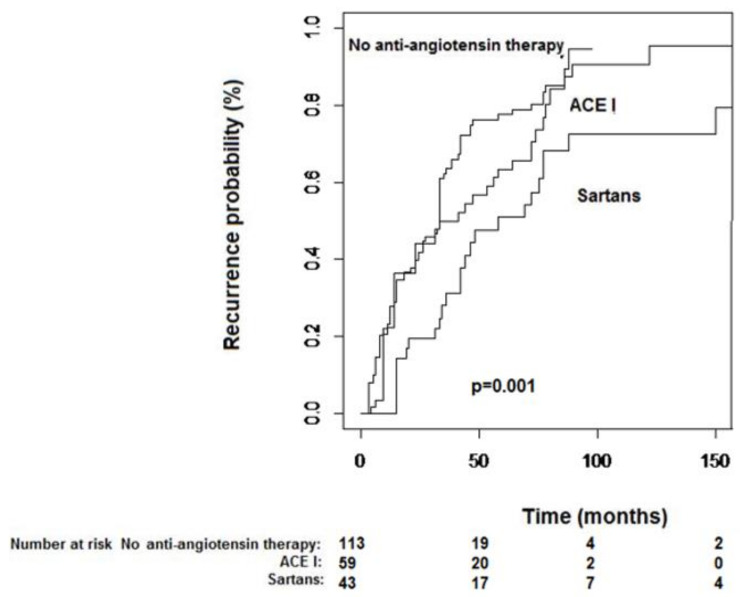
Kaplan–Meier curves of time-to-recurrence stratified by anti-hypertensive therapy.Median time-to-recurrence was 33 months (24–35) in group 1, 41 (23–72) in group 2 and 51 months (42–88) in group 3 (*p* = 0.001).

**Table 1 biomedicines-08-00399-t001:** Baseline characteristics of the study population.

Variable		Overall 215 pts	No Therapy 113 pts	ACE Inhibitors 59 pts	Sartans 43 pts	*P*
**Age:**	years	70 (39–86)	70 (48–86)	69 (39–86)	72 (48–82)	0.87
**Gender:**	male	171 (79.7%)	89 (79.5%)	45 (77.5%)	37 (83.9%)	0.78
**BMI:**	kg/m^2^	23 (17–36)	21(17–33)	24 (18–36)	24 (19–35)	0.45
**Mean arterial pressure:**	mmHg *	96.4 ± 15.5	87.6 ± 17.4	103.3 ± 19.4	100 ± 17.9	0.09
**Mellitus Diabetes**		81 (37.9%)	38 (34.2%)	24 (40.8%)	19 (42%)	0.66
**Etiology**						0.89
HCV		99 (46.4%)	48 (42.4%)	30 (51%)	21 (48.3%)	
HBV		46 (21.6%)	28 (24.6%)	11 (18.3%)	7 (19.3%)	
Other		70 (32%)	37 (33%)	18 (30.7%)	15 (32.4%)	
**Child-Pugh**						0.54
	A	184 (85.6%)	97 (86.3%)	48 (81.6%)	39 (90.3%)	
	B	31 (14.4%)	16 (13.7%)	11 (18.4%)	4 (9.7%)	
**MELD**		9 (6–17)	9 (6–17)	9 (7–15)	9 (7–13)	0.48
**Presence of Portal Hypertension +**		99 (46.4%)	45 (40%)	19 (32.6%)	31 (54.8%)	0.14
**Alpha-fetoprotein:** IU/mL		24 (1.1–2100)	20.4 (1.1–2100)	26.5 (1.7–1228)	37 (2.6–2100)	0.10
**BCLC**						0.88
0		22 (10.4%)	8 (9.5%)	7 (12.2%)	7 (9.6%)	
A		193 (89.6%)	105 (90.5%)	52 (87.8%)	36 (90.4%)	
**ALTSG**						0.88
I		22 (10.4%)	8 (9.5%)	7 (12.2%)	7 (9.6%)	
II		193 (89.6%)	105 (90.5%)	52 (87.8%)	36 (90.4%)	
**CLIP**						0.71
0		119 (55.5%)	61 (54.8%)	36 (61.2%)	22 (48.4%)	
1		84 (39.2%)	45 (39.7%)	19 (32.6%)	22 (48.4%)	
2		12 (5.3%)	7 (5.5%)	4 (6.2%)	1 (3.2%)	
**Number of nodules**		1 (1–3)	1 (1–3)	1 (1–2)	1 (1–3)	0.68
**Max Diameter:**	mm	30 (10–45)	29 (14–45)	30 (10–45)	30 (10–40)	0.92
**ECOG Performance Status:**	0	215 (100%)	113 (100%)	59 (100%)	43 (100%)	1.0

Categorical variables reported as frequency (percentage) and continuous variables reported as median (IQR) or mean ± standard deviation as appropriate. Comparisons were performed using Kruskal–Wallis test or Chi-square test when appropriate. Abbreviations: BMI, Body Mass Index; HCV, Hepatitis C Virus; HBV, Hepatitis B Virus; MELD, Model for End-Stage Liver Disease; BCLC, Barcelona Clinic Liver Cancer; ALTSG, American Liver Tumor Study Group; CLIP, Cancer of the Liver Italian Program; ECOG, Eastern Cooperative Oncology Group. * Computed as diastolic pressure + [1/3 × (systolic pressure − diastolic pressure)]. + Defined by at least one of the following: esophageal varices, platelet count <100,000/µL, and splenomegaly.

**Table 2 biomedicines-08-00399-t002:** Univariate/multivariate analysis of prognostic factors for overall survival.

Variable	Univariate Analysis	*P*-Value	Multivariate Analysis	*P*-Value
Age (reference ≤ 65 years)	**1.56 (1.01–2.42)**	**0.04**	1.89 (0.43–2.44)	0.21
Gender (reference Female)	0.96 (0.54–1.69)	0.89		
Blood hypertension (reference no)	1.08 (0.79–1.34)	0.89		
BMI (reference ≤ 25)	1.25 (0.84–1.42)	0.46		
Mellitus diabetes (reference no)	1.34 (0.95–1.68)	0.25		
Etiology (reference HBV)	HCV: 0.77 (0.46–1.41) Other:1.54 (0.77–3.09)	0.42		
Child-Pugh (reference A)	**2.83 (1.70–4.71)**	**<0.001**	**2.58 (1.44–4.62)**	**0.001**
Portal hypertension (reference no)	0.86 (0.56–1.30)	0.48		
AFP (reference ≤ 20 IU/mL)	**2.07 (1.89–3.5)**	**0.03**	1.71 (0.45–3.2)	0.17
MELD (reference ≤ 7)	**2.13 (1.21–3.75)**	**<0.001**	**2.37 (1.33–4.22)**	**0.003**
Max diameter (reference ≤ 30 mm)	1.12 (0.68–1.82)	0.65		
BCLC (reference 0)	1.54 (0.80–2.98)	0.19		
CLIP (reference 0)	1: 2.11 (1.8–3.3) 2: 3.5 (2.1–19.2)	0.07		
Number of nodules (reference 1)	>1: 1.75 (0.92–2.04)	0.25		
Anti-angiotensin therapy (reference none)	ACE I: 0.91 (0.76–2.10) Sartans: 0.71 (0.46–1.10)	0.15 0.39 * 0.12 *		

Reported as Hazard Ratio (95% CI). Significancies were reported in bold. Abbreviations: CI 95%, confidence interval 95%; BMI, Body Mass Index; HBV, hepatitis B virus; HCV, hepatitis C virus; AFP, alpha-fetoprotein; MELD, Model for End-Stage Liver Disease; BCLC, Barcelona Cancer of the Liver Clinic; CLIP, Cancer of the Liver Italian Program. * Correction by means of Bonferroni method.

**Table 3 biomedicines-08-00399-t003:** Univariate/Multivariate analysis of prognostic factors for time to recurrence.

Variable	Univariate Analysis	*P*-Value	Multivariate Analysis	*P*-Value
Age (reference ≤ 65 years)	**2.02 (1.27–3.21)**	**0.002**	1.59 (0.73–2.34)	0.21
Gender (reference female)	1.63 (0.89–3.01)	0.07		
Blood hypertension (reference no)	1.28 (0.82–1.74)	0.79		
BMI (reference ≤ 25)	1.34 (0.89–1.78)	0.21		
Mellitus diabetes (reference no)	1.46 (0.91–2.04)	0.16		
Etiology (reference HBV)	HCV: 0.92 (0.78–1.35) Other: 1.24 (0.87–3.89)	0.39		
Child-Pugh (reference A)	1.05 (0.57–1.92)	0.87		
Portal hypertension (reference no)	0.73 (0.47–1.14)	0.16		
AFP (reference ≤ 20 IU/mL)	1.10 (0.73–1.66)	0.62		
MELD (reference ≤ 7)	0.73 (0.43–1.24)	0.25		
Max diameter (reference ≤ 30 mm)	0.90 (0.54–1.51)	0.71		
BCLC (reference 0)	1.11 (0.59–2.09)	0.72		
CLIP (reference 0)	1: 1.51 (0.8–3.3) 2: 3.9 (0.91–7.2)	0.15		
Number of nodules (reference 1)	**2.45 (1.32–3.04)**	**0.02**	**1.45 (1.02–3.01)**	**0.04**
Anti-angiotensin therapy (reference none)	ACE I: 0.71 (0.46–1.12) Sartans: **0.49 (0.28–0.84)**	**0.001** 0.44 * **0.03 ***	0.78 (0.49–1.22) **0.47 (0.27–0.82)**	**0.008** 0.31 * **0.009 ***

Reported as Hazard Ratio (95% CI). Significancies were reported in bold. Abbreviations: CI 95%, confidence interval 95%; BMI, Body Mass Index; HBV, hepatitis B virus; HCV, hepatitis C virus; AFP, alpha-fetoprotein; MELD, Model for End-Stage Liver Disease; BCLC, Barcelona Cancer of the Liver Clinic; CLIP, Cancer of the Liver Italian Program. * Correction by means of Bonferroni method.

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
