# Peer review of "Angiotensin Receptor 1 Blockers Prolong Time to Recurrence after Radiofrequency Ablation in Hepatocellular Carcinoma patients: A Retrospective Study"

_biomedicines, 2020, doi:10.3390/biomedicines8100399_

Round 1

Reviewer 1 Report

The study investigated the correlation of angiotensin II type 1 receptor blocker treatment with recurrence-free and overall survival after radiofrequency ablation in hepatocellular carcinoma patients. It reports that sartan treatment was associated with increased recurrence-free survival, but not with increased overall survival.

It is a well-conducted study though, its significance is limited because of its retrospective, non-randomised study design. This reservation is mentioned in the Discussion, yet it must also be mentioned at least in the Abstract, if not in the title. For example, the latter could read: "Angiotensin receptor 1 blockers prolong time to recurrence after radiofrequency ablation in hepatocellular carcinoma patients: a retrospective study".

In addition, more emphasis must be placed on the fact that sartan treatment was not associated with increased overall survival. For example, the last sentence in the Abstract could read: "Sartans significantly improved time to recurrence after radiofrequency ablation in 31 hepatocellular carcinoma patients, but did not improve overall survival".

Please note that I am not an expert in statistics, and I thus do not have the authority to evaluate the statistical methods in a competent manner.

Author Response

The study investigated the correlation of angiotensin II type 1 receptor blocker treatment with recurrence-free and overall survival after radiofrequency ablation in hepatocellular carcinoma patients. It reports that sartan treatment was associated with increased recurrence-free survival, but not with increased overall survival.

It is a well-conducted study though, its significance is limited because of its retrospective, non-randomised study design. This reservation is mentioned in the Discussion, yet it must also be mentioned at least in the Abstract, if not in the title. For example, the latter could read: "Angiotensin receptor 1 blockers prolong time to recurrence after radiofrequency ablation in hepatocellular carcinoma patients: a retrospective study".

RE: The title was amended as suggested.

In addition, more emphasis must be placed on the fact that sartan treatment was not associated with increased overall survival. For example, the last sentence in the Abstract could read: "Sartans significantly improved time to recurrence after radiofrequency ablation in 31 hepatocellular carcinoma patients, but did not improve overall survival".

RE: The last sentence of the abstract was amended as suggested.

Please note that I am not an expert in statistics, and I thus do not have the authority to evaluate the statistical methods in a competent manner.

RE: The first author of the manuscript is an expert in statistics and performed the statistical analysis for more than 100 papers indexed in Pubmed. Thank you for your important and meaningful revision!

Reviewer 2 Report

The authors reported the effect of Sartans on the prevention of the HCC recurrence after RFA. The focus is interesting and their data are appropriately presented in the manuscript, therefore, it can be accepted in the current form. It is nice however, if the authors could discuss the potential mechanisms of this results regarding the prevention of the carcinogenesis not only in theliver but also in the other organs.

Author Response

The authors reported the effect of Sartans on the prevention of the HCC recurrence after RFA. The focus is interesting and their data are appropriately presented in the manuscript, therefore, it can be accepted in the current form. It is nice however, if the authors could discuss the potential mechanisms of this results regarding the prevention of the carcinogenesis not only in the liver but also in the other organs.

RE: Following the interesting reviewer’s comment, we added a further chapter in the Discussion commenting this aspect: “The aforementioned results might be of interest in oncology as sartans were found to determine interesting survival outcomes also in other kinds of cancers, for example renal carcinoma. A recent pooled analysis published by McKay et al considering 4,736 patients with metastatic renal cell carcinoma, of which 1,487 ACE-I/sartan users, demonstrated this positive associations [36].  This therapy has also shown positive clinical outcomes in advanced stages (e.g. metastatic lung cancer) and highly aggressive tumor types (e.g. glioblastoma and pancreatic ductal adenocarcinoma), as suggested in a recent study by Liu et al [37]. Similarly, the beneficial effects of sartans in cancer prevention were pointed out in several studies conducted in different kinds of malignancies. Two large epidemiological studies conducted in the UK [38] and in Taiwan [39] demonstrated a positive association between ACE-I/sartan use and a reduced risk of developing cancer, in particular colorectal cancer. Therefore, our study sheds further light on the potential anti-oncotic effects of sartans in the clinical practice.” Three new references were added to support these statements.